# A Personal Journey across Fluorescent Sensing and Logic Associated with Polymers of Various Kinds

**DOI:** 10.3390/polym11081351

**Published:** 2019-08-14

**Authors:** Chao-Yi Yao, Seiichi Uchiyama, A. Prasanna de Silva

**Affiliations:** 1School of Chemistry and Chemical Engineering, Queen’s University, BT9 5AG Belfast, Northern Ireland; 2Graduate School of Pharmaceutical Sciences, The University of Tokyo, 7-3-1 Hongo Bunkyo-ku, Tokyo 113-0033, Japan

**Keywords:** fluorescence, polymer, particle, sensor, logic gate, pH, ion, temperature

## Abstract

Our experiences concerning fluorescent molecular sensing and logic devices and their intersections with polymer science are the foci of this brief review. Proton-, metal ion- and polarity-responsive cases of these devices are placed in polymeric micro- or nano-environments, some of which involve phase separation. This leads to mapping of chemical species on the nanoscale. These devices also take advantage of thermal properties of some polymers in water in order to reincarnate themselves as thermometers. When the phase separation leads to particles, the latter can be labelled with identification tags based on molecular logic. Such particles also give rise to reusable sensors, although molecular-scale resolution is sacrificed in the process. Polymeric nano-environments also help to organize rather complex molecular logic systems from their simple components. Overall, our little experiences suggest that researchers in sensing and logic would benefit if they assimilate polymer concepts.

## 1. Introduction

It has been our pleasure to investigate molecular-scale devices which communicate with us at the human-scale. Owing to their subnanometric dimensions, they operate across a range of larger size-scales and provide us with valuable information from these worlds. Fluorescence signals provide output while various chemical species serve as input signals. Excitation light powers these devices wirelessly. In order to carry information, some modulation is required in the fluorescence signal. Chemical responsiveness provides this by chemically biasing a competition [1,2,3,4,5,6] for the deactivation of the fluorophore excited state between fluorescence emission [7,8] and photoinduced electron transfer (PET) [9,10]. Because of the extreme nature of this responsiveness, it is easy to regard these devices as ‘off-on’ switches. This leads to the realization that molecular devices share many attributes with semiconductor logic counterparts [11], while differing in other features [12,13,14,15,16,17,18,19,20,21,22,23,24,25,26,27,28,29,30,31,32,33,34,35,36,37,38]. Chemical responsiveness of fluorescence signals can also be arranged via ionic/dipolar influences on internal charge transfer (ICT) excited states [2,8,39]. Extreme versions of this behavior can be seen in benzofurazan fluorophores which again lead to ‘off-on’ switchability. Having a binary digital basis in electronic engineering does not preclude analog operations for exquisitely fine measurements. Similarly, the Boolean character of molecular switching devices still allows for the accurate measurement of tiny changes in the input signal, whether it be a chemical concentration or a physical property, when substantial populations of molecules exert mass action. Accurate sensing is therefore available from digital molecular devices. A significant fraction of our research involves polymers of some kind, sometimes in crucial ways.

Although each of the authors had their research formation in photoscience of small molecules [40,41,42,43], it is clear to us that macromolecules have uniquely beneficial characteristics barred to small counterparts [44]. For instance, polymer molecules are large enough to possess their own environments at the nanometer-scale. Although objects as varied as proteins [45] and DNA origami [46] could be studied in this way, it would be more immediately productive to pay attention to simple symmetrical systems such as quasi-spherical detergent micelles in water. We can consider detergent micelles in water as supramolecular polymer systems held together by hydrophobic interactions and then examine the region bounded by their surfaces for H^+^ distribution for instance. These are discussed in Section 2. Especially when cross-linked, polymer molecules are large enough to create their own phase-separated environments at the nanometer- to millimeter-scale. When solid particles are formed in this way, they serve as recyclable matrices to carry functional small molecules such as sensors. Section 3 represents these. Solid polymer particles can also be vehicles for functional small molecules such as drug candidates during their synthesis and their evaluation. These came to the fore during the combinatorial chemistry wave [47] and still have roles to play. It would be important therefore to be able to identify these vehicles individually within large populations. Section 4 presents a solution to this problem by tagging these vehicles with molecular logic gates. Linear macromolecules without cross-links can also create their own phase-separated environments in certain instances. Such a transition of extended linear to globular forms can occur as the temperature is ramped across a threshold value. Such transitions persist in some cross-linked gel versions as well. Fluorescence readout of these transitions is possible from polymer-linked probes. This opens the way to molecular thermometers, which are now throwing light on the foundations of biology (Section 5).

As indicated above, the polymer plays a variety of roles in these systems. These roles will depend on the chemical structures involved. Nano-environments will be set up by long hydrocarbon chain monomers carrying hydrophilic termini which aggregate in water. These micelles or membranes are non-covalent macromolecular (self-assembly) systems which are sisters of synthetic polymers. Some of these nano-environments will also be employed in an organizational role to assemble logic gates. Recyclable matrices will be created with diamondoid Si–O lattices. Vehicles for other molecules will be built from crosslinked polystyrene cores with oligoethyleneglycol shells. Sharp thermoresponsivity will be introduced with polyacrylamides carrying 2-propyl substituents and relatives.

## 2. Mapping Membrane-Bounded Species

Since compartmentalization is a key to the origin and maintenance of life, it is crucial to study membrane-bounded species, especially those which are key players in biology. H^+^ is paramount in this capacity because of its vital role in bioenergetics [48]. Since fluorescent PET signalling began with H^+^ sensing [49], Anthracenemethylamine derivative **1** (Scheme 1) is a straightforward adaptation of a ‘fluorophore-spacer-receptor’ system [50,51] with the addition of an anchoring module in the form of a hydrocarbon chain and a spatial tuning module in the form of amine substituents [52]. When H^+^ is picked up by **1** from its neighbourhood, the amine receptor is no longer able to perform a PET operation to the anthracene fluorophore, and the fluorescence is switched ‘on’. The neighbourhood being sampled is determined by the height/depth of the amine lone electron pair relative to the micelle-water interface, which in turn is controlled by the hydrophobicity of **1** as it gravitates to the appropriate point along the hydrophobicity/hydrophilicity continuum between polar water and the apolar micelle interior. The spatial tuning groups make fine adjustments to the positioning of the amine receptor. The local H^+^ density relative to the value in bulk water is related to the difference in p*K*_a_ values determined by fluorescence-pH titrations for **1** in micellar media and for a very hydrophilic version of **1** in neat water [53]. Such Δp*K*_a_ values obtained for structural variants of **1** can be correlated with the hydrophobicity of the spatial tuning module. These graphs provide a first glimpse into the spatial distribution of membrane-bounded H^+^ and how it is controlled by electrostatic and dielectric effects [53].

A more proper mapping of H^+^ in these micellar neighborhoods, in a cartographic sense, is achievable if the probe position can be determined at the same time as the Δp*K*_a_ measurement. This is made possible by employing a variant of **1** outfitted with a fluorophore whose emission wavelength is dependent on environmental polarity. The position occupied by the probe on the hydrophobicity/hydrophilicity continuum between polar water and the apolar micelle interior will reflect the local polarity experienced by the probe, and hence its emission wavelength. ICT fluorophores fit the bill [54,55,56,57], and benzofurazans [58,59,60,61,62,63] are the best of all in our hands. **2** (Scheme 1) and its close derivatives produce rather educational maps of H^+^ density near neutral Triton X-100 micelles [64] in water. In these fluorescent sensors, the effects of protonation at the terminal amino moiety (during H^+^ sensing) on the fluorophore are dominantly observed in fluorescence efficiency but not in original absorption and emission wavelengths of the fluorophore, which enables accurate monitoring of both H^+^ density and the environmental polarity simultaneously. As shown in Figure 1, the H^+^ density near Triton X-100 micelles is hardly affected until we approach neighborhoods of an effective dielectric constant (ε) 40. As sensors go towards the micellar interior from the position of ε = 40 to that of ε = 15, H^+^ density becomes suppressed to approximately 4% due to the dielectric repulsion (Figure 1). Our probes within the family represented by **2** are unable to get any closer to the micelle.

Further exploration of planet micelle is possible with **3** (Scheme 1) by providing useful mapping data from charged micelles. Although structurally close to **2**, **3** has no hydrogen-bond donor N-H group on the fluorophore. This is crucial because the N–H group at the anilino position is free from both protonation and deprotonation in a wide range of pH (e.g., 3 ≤ pH ≤ 12) in water or aqueous micellar solutions and thus can engage in multiple hydrogen bondings with anionic head-groups of micelles (e.g., sulfate groups with considerable hydrogen bonding ability in sodium dodecyl sulfate (SDS) micelles [65]) to pin the probe to a narrow location, meaning that detailed mapping was not possible for anionic micelles via only hydrophobicity tuning. Once N–H is replaced by N–CH_3_, this pinning effect disappears and a larger spatial distribution of probe positions opens up [66]. **2** does not fare much better with cationic micelles because of cation–pi interactions [67] between the micelle head-groups and the probe pi-system. **4** (Scheme 1) has stronger hydrophobic interactions due to the dioctyl chains so that the cation-pi interaction is relegated to a minor role. Better mapping is the result, though higher-resolution data remains our long-range goal.

Sentient beings depend on Na^+^ near nerve membranes to convey and process environmental signals [68]. Membrane-bounded Na^+^ is estimated by **5** (Scheme 2) [69], which takes a leaf out of **1′**s book by using a hydrocarbon chain for gross targeting and anchoring of the probe in the micelle. Owing to the relative structural complexity of the benzo-15-crown-5 ether receptor in **5** for Na^+^ vis-à-vis the H^+^ receptor amine in **1**, no spatial tuning module is available so far. Nevertheless, it is gratifying to find that Na^+^ is concentrated a 100-fold near the surface of anionic micelles, whereas Na^+^ is repelled so much from cationic micelles and even neutral micelles as to be immeasurable with **5**. The latter finding need not be a surprise because hydrophilic Na^+^ would indeed be difficult to accommodate in a hydrophobic micelle neighbourhood when bulk water is available within travelling distance.

Though nanometric in size, micelles are great containers which can organize sets of functional molecules. A pair of a fluorophore and a receptor is such an example of a self-assembled fluorescent PET sensor. Here, the role of the spacer in the fluorescent PET system is taken over by the micelle itself [70]. Inspired by this concept, we extend it to self-assembled AND logic systems with, e.g., a fluorophore and two different selective receptors [71]. The fact that various logic gates can be constructed by the step-by-step addition of components allows a ‘plug-and-play’ approach to some of the simpler molecular logic functions. Covalently bound AND logic gates operated within micelles represent computing at the smaller end of the nanoscale [72], which semiconductor devices still struggle to do.

We can cross from mapping and logic to the seemingly unrelated topic of photosynthetic reaction centre (PRC) mimics. In nature, the PRC is a marvel of supramolecular organization within a membrane in terms of structure and function [73]. This is a good thing too, since our origin and survival depend on it. We turn to micelles as a model membrane to contain PRC mimics of a receptor_1_-spacer_1_-fluorophore-spacer_2_-receptor_2_ format [74]. These have two PET pathways originating from the opposite termini of **6** (Scheme 2), of which one is favoured, somewhat similar to what is seen in the PRC. In its excited state, **6** has an internal electric field [75] to direct PET in one direction rather than the other.

## 3. Solid-Bound Sensors

Being modular, fluorescent sensors of the ‘fluorophore-spacer-receptor’ format are easily extended to ‘fluorophore-spacer-receptor-spacer-particle’ systems, e.g., **7** (Scheme 2) [76]. Beside the practical aspect of reusability, this SiO_2_-bound amine receptor system shows the retardation of PET compared to homogeneous solution counterparts. Charge-separating processes of this kind are naturally slowed at solid surfaces because charge-stabilizing orientation polarization of water dipoles is less likely. Fortunately, this PET process, even after retardation, remains competitive with the radiative rate and so adequate H^+^-sensing capability remains. This situation is maintained when H^+^-sensing PET systems like **8** (Scheme 3) are embedded in polyvinylchloride, provided the polymer is suitably plasticized [77], and when Na^+^-sensing PET systems like **9** (Scheme 3) are bonded to various fibres [78]. Solid-bound PET sensors continue to grow in number [79,80,81,82].

## 4. Molecular Computational Identification (MCID)

In the previous section, the emphasis was on the fluorescent function with the particle being the new environment. Now the particle takes centre stage with the fluorescent function serving as an ID tag. As trailed above, polymer particles can be vehicles for various functional molecules but they can also be models for biological cells showing the way to cell diagnostics.

Whenever we encounter populations of objects, individual object identification is not a problem if they are at fixed locations. If they are not spatially addressable, some kind of tracking feature becomes necessary. Metamorphosing objects would also need tracking if some sense is to be made of their population. Modern information-based society is full of radiofrequency ID (RFID) tags which serve this tracking need [83], but these are limited to sizes above 10 μm because of the necessary antenna. Micrometric objects such as polymer particles and cells are therefore untouched by the RFID revolution and remain at large. Molecules would be capable of rectifying this situation if they possessed some easily detectable parameter which comes in a sufficiently large number of distinguishable values. For example, excitation and emission wavelengths of various fluorophores can encode up to 100 polymer beads, but not much more [84]. However, what about larger populations? It is possible to amplify these 100 codes many-fold by taking each fluorophore and making it conditionally switchable [85]. A given logic type represents this light output driven by chemical input. Many single-, double- and higher-logic types are available [26], e.g., PASS 1 (**10**, H^+^-input), YES (**11**, H^+^-input), and AND (**12**, Na^+^-, H^+^-inputs) (Scheme 3). Ternary logic types can be included as well [86]. Further amplification of diversity is made by attaching two or more tags to a given particle (Figure 2) [85,87,88]. The fluorescence wavelengths can extend into the near infrared to offer additional bandwidth [88]. MCID has recently been applied to populations, albeit rather small, so that object-to-object variations can be quantified (Figure 2). The method can then be used to unambiguously divide a population into sub-populations of a given logic type [88].

The mechanism of switching in YES gate **11**, AND gate **12** and the YES logic-based examples in Figure 2 are PET processes occurring within ‘fluorophore-spacer-receptor-spacer-particle’ and related type systems. Here, PET originates from an electron donor amine or benzocrown ether and terminates in an anthracene or azaBODIPY fluorophore. The PET rate is controlled by its thermodynamics as well as the length of the spacer. As usual, the PET process is arrested by protonation of the amine or by binding Na^+^ to the benzocrown ether.

It is appropriate to mention some drawbacks, challenges and potential applications of MCID. The need to wash the samples with a chosen reagent can be considered as a drawback from some viewpoints, but chemical stimuli are common in biology. Applications of MCID can be imagined in tracking members of combinatorial chemistry libraries at the level of single polymer beads. The challenge will be to popularize this application. The road to application in cell diagnostics will be rockier, since MCID tags responding to suitably benign chemical stimuli would need to be found and validated.

## 5. Molecular Thermometers

As mentioned in the introduction section, some linear macromolecules switch between extended and collapsed forms in water as a response to varying temperature. Such cases with hydrophilic and hydrophobic groups which balance their effects display a lower critical solution temperature. The high degree of polymerization of these systems leads to strongly cooperative behaviour so that the transition occurs across a rather narrow temperature range. Outfitting of these polymers with a small amount of an environment (e.g., polarity and hydrogen bonding)-sensitive ICT fluorophore, 4-*N*,*N*-dimethylsulfamoyl-7-aminobenzofurazan (DBD), by means of copolymerization produces a fluorescent thermometer, **13** (Scheme 4), with greatly increased sensitivity (Figure 3) [89] over previous versions [90,91]. Other ICT fluorophores such as 7-aminocoumarin [92], BODIPY [93,94,95,96], dansylamine [97,98] and 4-amino-7-nitrobenzofurazan (NBD) [99,100] have also been applied in a similar way. At lower temperatures, **13** takes an open and extended form, and the solvent water molecules access the fluorophores in **13** to cause quenching. In contrast, **13** exists in a globular form at a higher temperature, where the fluorophores are surrounded by the hydrophobic backbone, and therefore emits strong fluorescence. In the similar system **14** (Scheme 4)**,** fluorescence lifetime increases with temperature [101]. In contrast to fluorescence intensity, fluorescence lifetime is not influenced by the fluctuation of various experimental conditions (e.g., excitation light intensity and concentration of a sensor). Accordingly, the fluorescence lifetime can be a more reliable variable than the fluorescence intensity in some applications such as intracellular thermometry where the experimental conditions are relatively changeable. The downside is the need for more elaborate instrumentation for lifetime measurements.

In addition to high sensitivity, the polymeric design brings functional diversity to fluorescent thermometry. The functional temperature range can be easily tuned by using substituted monomers, e.g., **15** (Scheme 4) [89]. Interestingly, the functional temperature range of copolymer **16** (Scheme 4) with a 1:1 blend of the monomer units in **13** and **15** bisects the ranges of **13** and **15** [102]. So, the functional temperature range can be finely tuned by varying co-monomer feed ratios (Figure 3) [102]. The fluorescence wavelength of the polymeric thermometers can also be modified by using a different fluorophore, e.g., **17** (Scheme 4) [103] and **18** (Scheme 5) [104].

Using an additional component in copolymers can improve physical and chemical features of fluorescent thermometers. **19** (Scheme 5) is a highly water-soluble thermometer because the ionic 3-sulfopropyl acrylate units prevent interpolymeric aggregation [105]. Such high solubility enables highly-resolved temperature measurements. In contrast, the incorporation of an H^+^ receptor amine into a copolymer produces a molecular logic system with temperature and H^+^ as multiple inputs [106,107,108]. For instance, the polymeric logic gate **20** (Scheme 5) fluoresces strongly only when environmental H^+^ concentration is ‘low’ and temperature is ‘high’ to behave as an INHIBIT gate. High H^+^ levels protonate the receptor, causing the copolymer to adopt the extended structure. So the fluorophore is surrounded by water to cause quenching whatever the temperature.

The gelation of solution-based fluorescent polymeric thermometers by adding a crosslinker results in robust nanogel particles [109]. The representative gel **21** (Scheme 5) exists as nanometric beads and shows nearly an order-of-magnitude fluorescence enhancement over a small temperature range when dispersed in water. The switching mechanism is essentially the same as that seen with the soluble version **13** [89]. The thermo-responsive fluorescent polymeric structure based on **13** has also been used as a shell structure of multifunctional magnetite nanoparticles [110]. These particles are expected to be useful in anticancer heat treatment where monitoring the temperature of target tumours is important.

While these fluorescent polymeric thermometers enabled temperature measurements of small objects such as an aqueous fluid in a heater-equipped microdevice [100,111] and turbid aqueous media heated with ultrasound-irradiation [112], the most attractive target is certainly biological cells. Highly water-soluble fluorescent gels (**22**, Scheme 6) microinjected into monkey’s kidney COS7 cells showed a temperature-dependent fluorescence signal therein [113]. A linear polymeric thermometer (**23**, Scheme 6) unveiled temperature distribution of a COS7 cell with the aid of fluorescence lifetime imaging microscopy (FLIM), in which the inside of nuclei and neighbourhoods of mitochondria and centrosomes were remarkably hotter than the other cytoplasmic spaces (Figure 4) [114].

The above breakthrough in intracellular thermometry made biologists imagine the importance of the temperature at the single cell level and subsequently demand chemists to develop more user-friendly fluorescent thermometers. A cationic version of linear polymeric thermometers (**24**, Scheme 6) [115,116,117] now avoids microinjection procedures due to its ability to spontaneously enter cells. In addition, further labelling of this by a BODIPY structure (**25**, Scheme 6) enabled ratiometric thermometry, which offered high accuracy even without an expensive fluorescence lifetime imaging microscope [118]. Thermogenesis of brown adipocytes [119,120] and chemical stimulation of brain tissue [121] were successfully monitored with these thermometers.

The future of fluorescent polymeric thermometers looks quite bright. The exemplified application of fluorescent polymeric thermometers in brain tissue will accelerate their use in in-vivo thermometry beyond a single cell level. In such cases, NIR (near infrared) ICT fluorophores will be preferred for the deep penetration of an excitation light and return of an emission signal to a detector. Another challenge of fluorescent polymeric thermometers is targeting to specific organelles of live cells by incorporating target signals into their chemical structures. It is expected that the localization of the fluorescent polymeric thermometer in heat-generating organelles, e.g., mitochondria, improves the detectability towards intracellular thermogenesis [122]. The cytotoxicity of fluorescent polymeric thermometers has also been concerning in biological and medical studies, and normal cell division and even differentiation were observed in a recent progress [123].

## 6. Conclusions

Whether constructed covalently from monomers or not, whether aqueous soluble or not, whether cross-linked or not, or whether solid or not, polymers offer unique environments and objects as playgrounds for sensors and logic designers. These efforts lead to reusable sensors, insights into the spatial distribution of chemical species near interfaces, membrane-assembled logic systems, temperature maps within living cells, and identification protocols for submillimetric objects.

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
