# Peer review of "A Personal Journey across Fluorescent Sensing and Logic Associated with Polymers of Various Kinds"

_polymers, 2019, doi:10.3390/polym11081351_

Round 1

Reviewer 1 Report

Authors reviewed the current studies of fluorescent sensing and logic devices associated with polymers, which is interesting and helpful for this research filed. I would like to recommend its publication after addressing the following questions.

1.     In Section 2, the authors only summarized the fluorescence sensing mechanisms and molecular structures for hydrogen ions and metal ions sensing. The authors are suggested to discuss the role of polymer in this systems in detail.

2.     In Section 4,regarding about the polymers-based logic device, some drawbacks, challenges and potential applications should also be discussed.

3.     In Section 5, challenges and outlook of fluorescent polymeric thermometers need to be provided.

Reviewer 2 Report

Comments attached as a PDF file.

Reviewer 3 Report

Yao et al. have provided their expertise on stimuli-responsive polymer materials in terms of their functions and applications. The review is well-organized and provides the advancement of the field spanning over the last two decades. Furthermore, the role of functional groups in the polymers and their applications were quite explicit. 

However, a few minor points are suggested:

Page 3, line 133: due to its double chain--could be more specific (dioctyl group)

Page 5, line 205: MCID--needs to be defined before using it in the abbreviated form Or Page 4, line 184 should be abbreviated.

Page 3: In Compound 1 and 5, the octyl groups are skewed abruptly.

Page 3, line 96: starts to grow until at local ε of 15 the H+ density--sentence is not clear.

Page 2, line 93: 2 and close derivatives produce (2 and its close derivatives produce); Page 4, line 154: need (needs) etc.
